# Effect and Mechanism of Theaflavins on Fluoride Transport and Absorption in Caco-2 Cells

**DOI:** 10.3390/foods12071487

**Published:** 2023-04-01

**Authors:** Yueqin Fan, Zhendong Lei, Jiasheng Huang, Dan Su, Dejiang Ni, Yuqiong Chen

**Affiliations:** 1National Key Laboratory for Germplasm Innovation and Utilization for Fruit and Vegetable Horticultural Crops, Wuhan 430070, China; 2College of Horticulture & Forestry Sciences, Huazhong Agricultural University, Wuhan 430070, China

**Keywords:** theaflavins, fluoride, caco-2 cells, transport and absorption, mechanism

## Abstract

This paper investigated the effect and mechanism of theaflavins (TFs) on fluoride (F^−^) uptake and transport in the Caco-2 cell model through structural chemistry and transcriptome analysis. The results showed that the four major TFs (TF, TF3G, TF3′G and TFDG) at a 150 μg/mL concentration could all significantly decrease F^−^ transport in Caco-2 cells after 2 h of treatment and, at 2 μg/mL F^−^ concentration, the F^−^ transport was more inclined to efflux. During transport, the F^−^ retention in Caco-2 cells was significantly increased by TF3G while it was clearly decreased by TF. The interaction between TFs and F^−^ was analyzed by Raman spectroscopy and isothermal titration calorimetry, and F^−^ was shown to affect the π bond vibration on the benzene ring of TFs, thus influencing their stability. Additionally, F^−^ showed weak binding to TF3G, TF3′G and TFDG, which may inhibit F^−^ transport and absorption in the Caco-2 cell line. Transcriptome and RT-PCR analysis identified three key differentially expressed genes related to cell permeability, and TFs can be assumed to mediate F^−^ transport by regulating the expression of permeability-related genes to change cell monolayer permeability and enhance cell barrier function; however, this needs to be further elucidated in future studies.

## 1. Introduction

Tea, a natural beverage made from fresh leaves of tea trees (*Camelia sinensis* L.), is known as one of the world’s three major alcohol-free beverages because of its nutritional and health functions.

Fluoride (F^−^) is a trace element required to maintain the normal physiological function and life activities of the human body, and appropriate F^−^ intake is helpful to prevent the occurrence of dental caries and maintain the normal metabolism of calcium and phosphorus in the body [1]. Therefore, it is recommended that all ages over 8 years consume 3–4 mg of fluoride per day [2]. However, excessive fluoride intake will increase the risk of fluorosis, leading to dental fluorosis, skeletal fluorosis and other diseases [3]. The content of fluoride in dark tea (e.g., Qingzhuan tea) was investigated to be higher than in green tea and black tea, which may lead to public concern [4], because we know so little about the F^−^ bioavailability in tea. Studies have shown that F^−^ bioavailability can be affected by many factors. For instance, Sudan et al. found that among the different fluoride-containing components extracted from dark tea (Qingzhuan tea) in the Caco-2 cell model, the F^−^ transport and absorption was lower when tightly bound to polysaccharides than when loosely bound to polysaccharides [5]. Additionally, Zhang et al. used the same raw material to process green tea (with the highest polyphenol retention), black tea (with medium polyphenol retention) and dark tea (with the least polyphenol retention), and the F^−^ absorption in the three tea extracts was found to vary significantly in the Caco-2 cell model, with the largest F^−^ transport and absorption found in the green tea extract, followed by black tea extract, and dark tea extract [6]. These reports suggest that polyphenol oxides in tea may affect F^−^ absorption and transport, but few studies have been reported to date.

Theaflavins (TFs), important oxidation products of tea polyphenols, are abundant in black tea, dark tea and oolong tea, contributing to the yellow color and strong taste of tea infusion, and accounting for 2–6% of the dry weight of the aqueous extract in black tea [7]. At present, 25 kinds of theaflavins have been isolated and identified, with the main four components of TFs including theaflavin (TF), theaflavin-3-gallate (TF3G), theaflavin-3′-gallate (TF3′G) and theaflavin-3-3′-gallate (TFDG), whose structural formulas are shown in Figure 1 [8]. Theaflavins are characterized by easy electron loss and oxidation, due to the presence of multiple phenolic hydroxyl groups. Studies have found that theaflavins have antioxidant [9], anti-tumor [10], anti-inflammatory [11], antibacterial [12], antiviral [13], and anti-metabolic disease [14] properties, as well as other functions. In Ha-Young Park’s study, TFs were found to reduce the transport and absorption of fluorescent dyes in Caco-2 cells, increase the expression of TJ-related proteins and activate AMPK phosphorylation, thereby improving the intestinal barrier function [15].

The small intestine is the main location of human digestion and the absorption of substances, and the human colon adenocarcinoma (Caco-2) cell model is a classic model for simulating the intestinal environment in vitro and studying the absorption and metabolism of substances [16]. Caco-2 cells can spontaneously undergo epithelioid differentiation under in vitro culture conditions to form a monolayer of tightly connected cells with an intestinal lumen side (apical villus surface) and an intestinal wall side (basal surface), which possess the morphology and typical functions of the intestine. [17].

As there are few studies on the effects of tea polyphenol oxides on fluoride transport and absorption, this study aimed to investigate the effects and the related mechanisms of four major theaflavins (TF, TF3G, TF3′G and TFDG) on F^−^ uptake and transport in Caco-2 cells through structural chemistry and transcriptome analysis, which may provide a theoretical basis for the new utilization of theaflavins.

## 2. Materials and Methods

### 2.1. Cell Culture

Caco-2 cells (Procell Life Science & Technology CO., Ltd., Procell CL-0050, Wuhan, China) were cultured in Dulbecco’s Modified Eagle Medium–High Glucose (DMEM, Gibco, Waltham, MA, USA), supplemented with 10% Fetal Bovine Serum (FBS, Gibco, USA), 1% Penicillin-Streptomycin (Gibco, USA), 1% MEM NEAA (BI, Kibbutz Beit Haemek, Israel), L-glutamine (Biofroxx, Einhausen, Germany) and 0.2% Mycoplasma Prevention Reagent (Yeasen Biotechnology Co., Ltd., Shanghai, China). Cells were cultured in monolayers at 37 °C in a humidified atmosphere of 5% CO_2_, with the medium renewed two or three times a week.

### 2.2. Model Evaluation

In order to observe cell monolayer morphology, caco-2 cells were seeded in 12-well transwells (10^5^ cells a well) and observed under an inverted microscope every two days. Cell models were evaluated by trans-epithelial electrical resistance (TEER) and alkaline phosphatase (AKP) activity. TEER was measured with a cell-resistance meter (Millipore, MERS00002, Burlington, MA, USA), and the TEER value > 400 Ω·cm^2^ indicated that Caco-2 cells formed a tightly connected monolayer and could be used for trans-membrane transport experiments. AKP activity was determined by the alkaline phosphatase assay kit as instructed by the manufacturer (Nanjing Jiancheng Bioengineering Institute, A059-2, Nanjing, China), and the ratio of AP side/BL side of alkaline phosphatase activity > 5 was defined as an indicator that the cell monolayer had formed polar differentiation. After 21 days of cell culture, the TEER was greater than 400 Ω·cm^2^ and AKP ratio was 5.79, indicating the cells could be used for trans-membrane transport experiments.

### 2.3. CCK-8 Viability Test

The CCK-8 assay was performed on Caco-2 cells to evaluate the cell viability in the presence of working solutions (TFs + DMEM, F^−^ + DMEM or TFs + F^−^ +DMEM working solution; TFs: Shanghai yuanye Bio-Technology Co., Ltd., Shanghai, China; F^−^(NaF): AR, Shanghai yuanye Bio-Technology Co., Ltd., Shanghai, China). Briefly, Caco-2 cells were seeded in 96-well plates (10^4^ cells a well) for 3–4 d to confluence of 80–90% in the bottom of the wells and supplemented with different working solutions. After 24 h incubation with different tested drugs, the medium was removed, followed by the addition of 110 μL of CCK-8 solution (10 µL CCK-8 reagent plus 100 µL complete DMEM) and incubation for 1 h at 37 °C. Finally, the absorbance at 450 nm was measured with a microplate reader. Six biological replicates were performed for each treatment.

### 2.4. Fluoride (F^−^) Transport Experiment

The fluoride transport experiment included forward transport test (AP-BL) and reverse transport test (BL-AP). In brief, the Caco-2 cells were seeded in 12-well trans-well plates (10^5^ cells a well) and cultured for 21 days until full confluence. After discarding the medium, the cells were washed 2–3 times with HBSS (Gibco, USA) and placed in the incubator after adding HBSS (pH = 6.0). After equilibration for 30 min, HBSS was discarded and 0.5 mL of co-transfer solution was added to the AP side (HBSS with different drug treatments, pH = 6.0) and 1.5 mL of HBSS was added to the BL side (vice versa for the reverse transport test), respectively. After incubation for the indicated time points, F^−^ content on the BL side was determined (the F^−^ content in the AP side was determined in reverse transport test). Time gradient transport experiment was conducted as follows: the TFs in the co-transfer solution were set at 150 μg/mL and the F^−^ concentration was set at 5 μg/mL. The treatment time was set separately for 0.5 h, 1 h, 2 h, 3 h and 4 h. TFs concentration gradient transport experiment was carried out as follows: TFs in the co-transfer solution were set at 25 μg/mL, 50 μg/mL, 100 μg/mL and 150 μg/mL, and the F^−^ concentration was set at 5 μg/mL for 2 h. F^−^ concentration gradient transport experiment was carried out as follows: F^−^ concentrations in the co-transfer solution were set at 0.2 μg/mL, 1 μg/mL, 2 μg/mL and 5 μg/mL, and the TFs concentration was set at 150 μg/mL for 2 h. Three biological replicates were performed for each treatment.

### 2.5. Effects of TFs on F^−^ Uptake by Caco-2 Cells

In briefy Caco-2 cells were seeded in 6-well plates (10^4^ cells a well) and cultured for 21 days, followed by discarding the medium and washing the cells 2–3 times with pre-warmed (37 °C) HBSS (pH 6.0). After equilibration in the incubator for 30 min with HBSS (pH 6.0), the culture solution was discarded, followed by the addition of 2.5 mL of co-transfer solution (150 μg/mL TFs + 5 μg/mL F^−^) to each well and incubation in the incubator for 2 h. After discarding the solution, the cells were washed 2–3 times with pre-cooled (4 °C) PBS buffer, followed by adding 500 μL cell lysis solution (PBS and RIPA mixed at 1:1 ratio) to each well, incubation on ice for 10 min and then centrifugation at 4 °C for 30 min at 12,000 rpm. Finally, total cell protein was detected using the BCA protein assay kit as instructed by the manufacturer (Shanghai Biyuntian Biotechnology Co., Ltd., Shanghai, China). F^−^ content was determined by F^−^ ion selective electrode. Three biological replicates were performed for each treatment.

### 2.6. Fluoride Ion Fluorescence Probe Localization Characterization

In brief, the Caco-2 cells were seeded in four-chamber confocal cell culture dishes (10^4^ cells a chamber), cultured to a dense monolayer, rinsed with pre-warmed (37 °C) PBS buffer, and incubated with 600 μL Probe1 [18] for 30 min. After rinsing with pre-warmed (37 °C) PBS, the cells were added to 150 μg/mL TFs + 2 μg/mL F^−^ (with PBS) or 2 μg/mL F^−^ (with PBS) working solutions, respectively. After incubation in the incubator for 30 min, cell fluorescence was observed using a laser sheet scanning confocal microscope (Leica Microsystems, TCS SP8 DLS, Wetzlar, Germany; excitation wavelength 552 nm).

### 2.7. Interaction and Structural Characterization of TFs with F^−^

In brief, TFs (with water) (1000 μg/mL) were homogeneously and separately mixed with 10, 100 and 1000 μg/mL of F^−^ for half an hour at room temperature. Next, the mixture was added to the flower-shaped silver substrate in small amounts, followed by vacuum-oven-drying at 60 °C for 0.5 h, and the Raman spectra were measured, with 10 repetitions for each sample (Shanghai Ruhai Photoelectric Technology Co., Ltd., Shanghai, China). The interaction between TFs and F^−^ was determined by isothermal titration calorimetry, with 5 mol/L TFs solution (with PBS) being titrated by 5 mol/L F^−^ solution (with PBS). The control was a 5 mol/L F^−^ solution titrated with the PBS solution.

### 2.8. Transcriptome Sequencing and Differentially Expressed Genes (DEGs) qRT-PCR Analysis

Cell culture and incubation followed the same procedures as described in 2.5. After adding pre-cooled TRIzol lysis solution (1 mL a well), the cell suspension was transferred to RNase-free cell lyophilization tubes and stored at −80 °C. RNA integrity was assessed using the RNA Nano 6000 Assay Kit of the Bioanalyzer 2100 system (Agilent Technologies, Santa Clara, CA, USA). Total RNA was used as input material for the RNA sample preparation. In brief, mRNA was purified from total RNA using poly-T oligo-attached magnetic beads. Fragmentation was carried out using divalent cations under an elevated temperature in First Strand Synthesis Reaction Buffer (5X). First-strand cDNA was synthesized using random hexamer primer and M-MuLV Reverse Transcriptase (RNase H-), and second-strand cDNA synthesis was performed using DNA Polymerase I and RNase H. Remaining overhangs were converted into blunt ends via exonuclease/polymerase activities. After adenylation of 3′ ends of DNA fragments, an adaptor with hairpin loop structure was ligated to prepare for hybridization. In order to preferentially select cDNA fragments of 370~420 bp in length, the library fragments were purified with AMPure XP system (Beckman Coulter, Brea, CA, USA). Then PCR was performed with Phusion High-Fidelity DNA polymerase, Universal PCR primers and Index (X) Primer. Finally, PCR products were purified (AMPure XP system) and library quality was assessed on the Agilent Bioanalyzer 2100 system. The clustering of the index-coded samples was performed on a cBot Cluster Generation System using TruSeq PE Cluster Kit v3-cBot-HS (Illumia) as instructed by the manufacturer. After cluster generation, the library preparations were sequenced on an Illumina Novaseq platform and 150 bp paired-end reads were generated. Three biological replicates were performed for each treatment.

The quantitative real-time fluorescence (qRT-PCR) was performed using the reverse transcription kit (goldenstar RT6 cDNA synthesis mix). The qRT-PCR amplification was performed using 2 × T5 Fast qPCR mix (SYBR green I). Relative expression of DEGs was calculated using the 2^△△CT^ method, using β-actin as the reference gene. The primers of DEGs were synthetized by Beijing Qingke Biotechnology Co., Ltd. (Beijing, China), and the primer sequences were shown in Appendix A.

### 2.9. Statistical Analysis

The data are presented as the mean ± standard deviation. Analysis was performed using IBM SPSS Statistics 25 and Microsoft Excel 2016. GraphPad Prism 9 and Origin2022 were used for plotting. Differences between groups were analyzed using one-way ANOVA, followed by least significant difference (LSD) multiple comparisons, with *p* < 0.05 considered significantly different.

## 3. Results

### 3.1. Effects of Test Substances on Viability of Caco-2 Cell

The effects of different concentrations of TFs and TFs + F^−^ on the viability of Caco-2 cells were analyzed and the results are shown in Figure 2. In Figure 2A, the inhibitory effect of TFs on Caco-2 cell viability was seen to increase with the increase in TFs concentration in the range of 50~500 μg/mL. At a TFs concentration lower than 200 μg/mL, the cell viability was higher than 80%, suggesting the cytotoxicity was small. Sodium fluoride (NaF) was more toxic to cells, and when the F^−^ concentration was lower than 50 μg/mL, the viability of Caco-2 cells was higher than 80% (Figure 2B). Under the treatment of 150 μg/mL TFs + 5 μg/mL F^−^, the viability of Caco-2 cells could reach more than 80% (Figure 2C), indicating that this combination has low cytotoxicity and can be used as a reference concentration in subsequent transport experiments.

### 3.2. Regulation of TFs on F^−^ Transport in Caco-2 Cells

#### 3.2.1. Effects of TFs on F^−^ Transport in Caco-2 Cells

Caco-2 cells cultured for 21 days were treated with NaF. With the increase in treatment time, the amount of F^−^ influx and efflux showed an increasing trend (Figure 3A,B), with a faster growth rate at 0 h~3 h, followed by a plateau. After 2 h, the influx and efflux of F^−^ were significantly lower in the TFs group than in the CK group (F^−^), indicating that TFs had a significant inhibitory effect on F^−^ transport after 2 h of treatment. Among all the TFs groups, TF3G had the strongest inhibitory effect on the influx transport of F^−^, followed by TF3′G.

The apparent permeability coefficient (Papp) of a drug can reflect its absorption rate in the human small intestine; thus, it can serve as an indicator for the bioavailability of a drug. It is generally believed that the intestinal absorption rate is 0~20% (poor absorption) at the Papp value of less than 1 × 10^−6^ cm/s, 20~70% (moderate absorption) at the Papp value between 1 × 10^−6^ cm/s and 10 × 10^−6^ cm/s, and 70~100% (high absorption) at the Papp value greater than 10 × 10^−6^ cm/s. The experimental results are shown in Table 1, where the influx and efflux apparent permeability coefficient Papp of F^−^ was seen to be greater than 10 × 10^−6^ cm/s under different treatment time points, indicating satisfactory F^−^ absorption. The Papp value increased within 0.5 h, but significantly decreased after 2 h, with the most significant drop being seen in TF3G treatment, indicating that TFs can reduce the bioavailability of F^−^ to a varying degree.

The apparent permeability coefficient ratio (P-ratio: Papp (BL-AP)/Papp (AP-BL)) can reflect the transport and absorption of drugs. A P-ratio value of greater than 1.5 is considered active drug transport while a P-ratio of less than 1.5 is mainly considered passive drug transport [19]. The experimental results showed that the P-ratio value was about 1 in the NaF treatment alone for a different period of time, indicating the dominance of passive NaF transport. After adding TFs for 1h, the P-ratio value was greater than 1.5 for TF3G and TF3′G and less than 1.5 for TF and TFDG, indicating that TFs had a significant effect on the transport mode of F^−^.

The effects of different concentrations of TFs on F^−^ transport in Caco-2 cells are shown in Figure 3C,D, where the inhibitory effect of TFs on F^−^ transport was seen to be enhanced with the increase in TFs concentration. Under the treatments of 25 μg/mL TF3′G, 50 μg/mL TF3G, 100 μg/mL TF, and 150 μg/mL TFDG, the F^−^ transport amount was significantly reduced. Meanwhile, TF3′G significantly inhibited the efflux transport of F^−^ at all treatment concentrations. At the TFs concentration of 150 μg/mL, the both-way transport of F^−^ was significantly lower in each TFs group than in the CK group (F^−^).

In Table 2, the Papp of each treatment group was higher than 10 × 10^−6^ cm/s, indicating satisfactory F^−^ absorption. At the TFs concentrations of 100 μg/mL and 150 μg/mL, the influx and efflux Papp were lower in the four TFs groups than in the (F^−^) control, indicating that the bioavailability of F^−^ could be reduced at certain concentrations of TFs. In addition, the P-ratio values were less than 1.5 for each group, indicating that F^−^ was mainly transported by passive transport, and different concentrations of TFs had little effect on the transport mode of F^−^.

In Figure 3E,F, the F^−^ transport rate was seen to decrease with the increase in F^−^ concentration in the range from 0.2 μg/mL to 2 μg/mL, but significantly increased at 5 μg/mL. Adding TFs could significantly reduce the influx transport of F^−^. The efflux transport rate of F^−^ was not significantly different in the F^−^ concentration range from 0.2 μg/mL to 2 μg/mL, but significantly increased at the F^−^ concentration of 5 μg/mL. Adding TF3′G and TFDG could significantly reduce the efflux transport rate of F^−^, while TF and TF3G had no obvious effect on the efflux transport rate of F^−^ under the condition of a low F^−^ concentration.

In Table 3, it was shown that TFs could significantly reduce the influx and efflux F^−^ Papp at different F^−^ concentrations, except for the efflux Papp of TF and TF3G at a 0.2 μg/mL F^−^ concentration.

#### 3.2.2. Effects of TFs on F^−^ Accumulation in Caco-2 Cells

The uptake amount of F^−^ in Caco-2 cells is shown in Figure 4, where the TF3G treatment group was seen to have the largest accumulation of F^−^ under the treatment of 150 μg/mL TFs + 2 μg/mL F^−^ for 2 h, which was significantly higher (*p* < 0.05) than that of the other groups, reaching 190.56 μg/g protein. Meanwhile, the TF treatment group showed the smallest F^−^ accumulation (139.51 μg/g protein), significantly lower (*p* < 0.05) than that of the other groups, indicating that different TFs treatments varied in their F^−^ accumulation.

#### 3.2.3. Distribution Characteristics of F^−^ in Caco-2 Cells

The localization of F^−^ in Caco-2 cells was characterized by the F^−^ fluorescent probe. In Figure 5, F^−^ was seen to be dispersed in Caco-2 cells and mainly distributed near the cell membrane. The cell boundary of the fluorescence field was the most obvious in the TF3G group, while it was most blurred in the TF group, similar to the CK group. The fluorescence intensity was calculated by Image J software (Figure 5), and the average fluorescence intensity was seen to be the highest in TF3G group and the lowest in TF group, consistent with the rule of F^−^ uptake amount in Caco-2 cells.

### 3.3. Interaction and Structural Characterization of TFs and F^−^

Polyphenols easily form complexes with metal ions and small molecules due to their phenolic hydroxyl structure and π-conjugated system.

In the Raman spectra before and after the interaction between TF and F^−^ (Figure 6A), TF itself was seen to have no attributable characteristic peak position, and after adding 10 μg/mL F^−^, the Raman spectrum of TF did not significantly change. With the increase in F^−^ concentration, the TF peak intensity significantly decreased, indicating that the high concentration of F^−^ was not conducive to TF stability, resulting in the decrease in its peak intensity.

In the TF3G Raman spectra (Figure 6B and Appendix A), the characteristic peaks at 637 cm^−1^, 940 cm^−1^, 1074 cm^−1^ and 1263 cm^−1^ are attributed to the ring deformation of mono-substituted benzenes, ring breathing, ring vibration of ortho-disubstituted benzenes and ring stretching of epoxy derivatives, respectively. After adding F^−^, the peak intensity of 1263 cm^−1^ increased, and two new characteristic peaks appeared at 1540 cm^−1^ and 1640 cm^−1^, which were attributed to the coupling C=C double bond stretching and C=C stretching, respectively, indicating that F^−^ may affect the structure of TF3G by influencing the π bond vibration on the benzene ring.

In the TF3′G Raman spectra (Figure 6C and Appendix A), the characteristic peaks at 640 cm^−1^, 714 cm^−1^, 1265 cm^−1^ and 1717 cm^−1^ are attributed to the ring deformation, C-C stretching of monosubstituted benzenes, ring stretching and C=O stretching of epoxy derivatives, respectively. The ring deformation of mono-substituted benzenes and the ring stretching of epoxy derivatives are the same as the TF3G spectra. At F^−^ concentrations of 10 μg/mL and 100 μg/mL, there was a certain red shift at 1309 cm^−1^ and 1603 cm^−1^, which were attributed to the in-plane deformation of CH_2_ and ring expansion, respectively, indicating that F^−^ mainly affected the partial epoxy structure of TF3′G.

The characteristic peaks of TFDG at 775 cm^−1^, 940 cm^−1^, 1259 cm^−1^, 1527 cm^−1^ and 1642 cm^−1^ are attributed to the ring vibration, ring breathing of para-disubstituted benzene, ring stretching, ring stretching and C=C stretching of epoxy derivatives, respectively. Meanwhile, F^−^ addition did not significantly change the peak position and peak intensity of the characteristic peaks (Figure 6D and Appendix A).

Isothermal titration calorimetry (ITC) is mainly used for quantitative analysis of the interaction between biological macromolecules and small molecules, and it can also be used to detect the interaction between small molecules. The principle is to monitor the heat changes in real time through microtitration and calculate the relevant thermodynamic parameters through the model, such as binding constant Ka, dissociation constant Kd, enthalpy change ΔH, entropy change ΔS, stoichiometric ratio n, Gibbs free energy ΔG etc. The titration heat spectrum and binding isotherm fitting diagram of F^−^and TFs are shown in Appendix A, and the thermodynamic simulation parameters are listed in Table 4, where the binding constant Ka of TF and F^−^ is seen to be 2.014 × 10^2^, and the stoichiometric ratio (n) is 0.023, which is close to 0, indicating that the two are not combined. Meanwhile, the binding constant Ka of TF3G and F^−^ is 1.854 × 10^3^, and the stoichiometric ratio (n) is 0.744, indicating a weak binding of TF3G with F^−^, while ΔH < 0 and -TΔS < 0 (Appendix A) indicated that the reaction between the two is driven by enthalpy and entropy. An enthalpy-driven reaction is usually generated by hydrogen bonds, ionic bonds, and van der Waals forces, and the entropy-driven process is usually attributed to hydrophobic interactions or conformational changes. The binding constant Ka of TF3′G and F^−^ is 3.262 × 10^1^, and the stoichiometric ratio (n) is 0.866, indicating a weak binding of TF3′G with F^−^, while ΔH > 0, -TΔS0, and -TΔS < 0 (Appendix A) indicated that the binding process is driven by entropy. As there was no conformational change in TFDG, hydrophobic interaction should have occurred between F^−^ and TFDG.

### 3.4. TFs Mediate F^−^ Absorption and Transport by Regulating the Expression of Cell Monolayer Permeability-Related Genes

The transmembrane transport of F^−^ is mainly carried out in a passive diffusion manner, generally through the cell bypass pathway [20]. Cell monolayer permeability is closely related to the transport of small molecules by the cell bypass pathway. GO functional enrichment and KEGG pathway enrichment analysis of DEGs between the treatment group and control group identified two GO nodes related to cell monolayer permeability (GO 0045216 of biological process (BP), which is related to the formation of cell–cell junction organization, and GO 0070160 of the cell component (CC), which is related to occluding junction) and one KEGG pathway related to cell monolayer permeability (hsa04530 pathway, which is associated with the tight junction of cells). A total of 103 DEGs were enriched in these three pathways. Based on the expression level and fold change, 3 DEGs related to cell–cell junction were screened out, including Wnt family member 11 (*WNT11*)*,* Tubulin alpha 1b (*TUBA1B*)*,* Actin gamma 1 *(ACTG1*).

The expression levels of the three key DEGs under different TFs treatments are shown in Figure 7A. Compared with the control group (CK), TF3G and TFDG showed a significant increase in the expression levels of *ACTG1*. Meanwhile, TF and TFDG treatments showed no change in the expression levels of *WNT11* and *TUBA1B*, in contrast to a significant decrease in their expression levels in TF3G and TF3′G treatments. These results suggest that the four different theaflavin monomers vary in their action mechanism and their inhibitory effect on F^−^ transport is based on the interaction of several genes.

The above experimental results revealed that TF3G had the most prominent inhibitory effect on F^−^ transport in the four theaflavin monomers, so the effects of different concentrations of TF3G on the expression of the three key genes for F^−^ transport were further analyzed (Figure 7B). Adding TF3G was seen to have a stimulative effect on the expression of *ACTG1*, with the strongest promotion effect at 50 μg/mL. Moreover, the expression levels of *WNT11* and *TUBA1B* were seen to decrease with the increase in TF3G concentration from 25 to 150 μg/mL.

The qRT-PCR analysis of the samples in TF3G and CK groups showed that the addition of TF3G down-regulated the expression of *WNT11* and *TUBA1B* while up-regulating the expression of *ACTG1* (Figure 7C), inferring that TF3G may regulate cell junction proteins and cell permeability by mediating the expression of these genes, thereby affecting F^−^ transport.

## 4. Discussion

Previous studies have revealed two major ways for small intestinal epithelial cells to transport drugs: (i) through transporters on the membrane of small intestinal epithelial cells, and (ii) by cell bypass through tight junctions between small intestinal epithelial cells [21]. Generally, lipophilic drugs are mainly transported by trans-membrane proteins, while hydrophilic drugs are mainly transported by the cell bypass. As a water-soluble ion, the F^−^ absorption and transport in the gastrointestinal tract are mainly performed by passive diffusion through the cell bypass [20].

TFs are polyhydroxy compounds, which can easily be combined with F^−^ by hydrogen bonds. Our experimental results revealed that the binding constant Ka and the stoichiometric ratio were 1.854 × 10^3^ and 0.744 for TF3G and F^−^, indicating a weak binding of TF3G with F^−^, while ΔH < 0 suggested that this reaction was driven by enthalpy and entropy. Enthalpy-driven processes are usually caused by the interaction of hydrogen bonds, ionic bonds, and van der Waals forces. Entropy-driven processes are usually generated due to hydrophobic interactions or conformational changes. Meanwhile, the binding constant Ka and the stoichiometric ratio were 3.262 × 10^1^ and 0.866 for TF3′G and F^−^, and 1.732 × 10^3^ and 0.645 for TFDG and F^−^, indicating that they also had weak binding with F^−^, while their ΔH > 0 and -TΔS < 0 suggested that the reaction was driven by entropy. Especially in the case of no conformational change in TFDG, hydrophobic interactions should have occurred. Therefore, the binding of F^−^ to TFs weakened F^−^ transport in Caco-2 cells by the cell bypass. As previously reported, TF3G, TF3′G and TFDG were regulated by efflux proteins such as P-glycoprotein and MRPs in Caco-2 cells, resulting in their poor transport and absorption and low bioavailability [22]. The binding of TF3G, TF3′G and TFDG to F^−^ caused a decrease in F^−^ transport in the Caco-2 cell model through transporters on the membrane of small intestinal epithelial cells.

TFs have an inhibitory effect on the transport of F^−^ in Caco-2 cells, which may also be related to the regulation of F^−^ transport by TFs by mediating the expression of genes related to cell monolayer permeability. *WNT11* plays an important role in the process of cell migration [23], and the decrease in *WNT11* expression (Figure 7C) may inhibit cell migration, thus strengthening the connection between cells. Changes in the expression levels of *ACTG1* and *TUBA1B* can adjust the cytoskeletal morphology, which may play an important role in the establishment of the cell junction [24]. The increased expression of *ACTG1* after the addition of TF3G may promote actin polymerization, thereby facilitating cell junction formation. In a study on the effect of theaflavins on Caco-2 cell membrane permeability by Park et al. [15], theaflavins were found to be able to increase the mRNA and protein expression of tight junction-related proteins (occludin, claudin-1 and ZO-1) in Caco-2 cells and enhance the phosphorylation of AMP-activated protein kinase, thereby enhancing intestinal barrier function.

In conclusion, the integrated results indicate that TFs could reduce F^−^ transmembrane passive transport by binding to F^−^, and TFs could also affect the permeability of cell monolayer by regulating the cell–cell junction in the intercellular space, thereby enhancing the intestinal barrier function and reducing F^−^ transport into the intestinal inner layer, achieving the inhibition of F^−^ absorption and utilization such as TF3G (Figure 8). The results show that it is defective to take only fluoride content of tea and fluoride leaching rate as the basis for human intake of fluoride during tea drinking, and the effectiveness of fluoride in tea should be taken into account. Our experimental results provide new insights into the absorption process of tea fluoride in human body.

## Figures and Tables

**Figure 1 foods-12-01487-f001:**
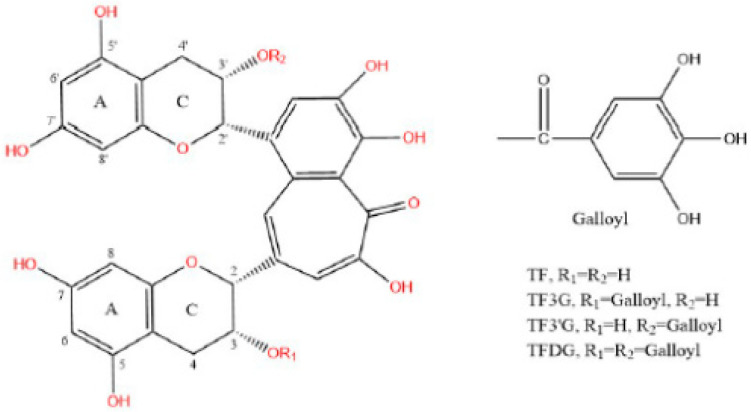
Structure of Theaflavins (TFs: TF, TF3G, TF3′G and TFDG).

**Figure 2 foods-12-01487-f002:**
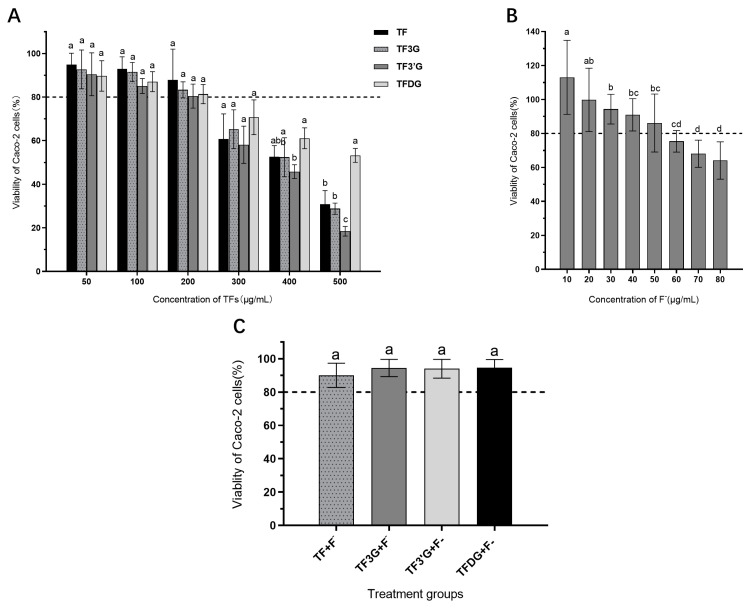
Effects of TFs and F^−^ treatment on the viability of Caco-2 cells. (**A**) Caco-2 cell viability treated with different concentrations of TFs for 24 h. (**B**) Caco-2 cell viability under different concentrations of F^−^ for 24 h. (**C**) Caco-2 cell viability treated with TFs + F^−^ mix for 4 h. Different lowercase letters in the same group indicate significant differences at *p* < 0.05.

**Figure 3 foods-12-01487-f003:**
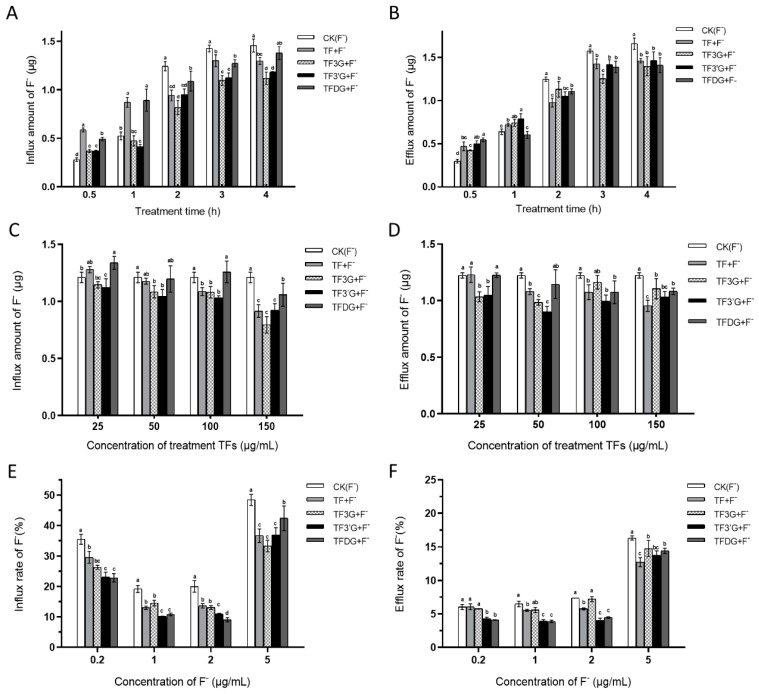
(**A**,**B**) Effect of TFs on F^−^ transport amount in Caco-2 cells under different treatment time. (**C**,**D**) Effects of different concentrations of TFs on F^−^ transport amount in Caco-2 cells. (**E**,**F**) Transport rate of F^−^ in Caco-2 cells under different F^−^ concentrations. (**A**,**C**,**E**) influx transport (AP-BL); (**B**,**D**,**F**) efflux transport (BL-AP). Different lowercase letters in the same group indicate significant differences at *p* < 0.05.

**Figure 4 foods-12-01487-f004:**
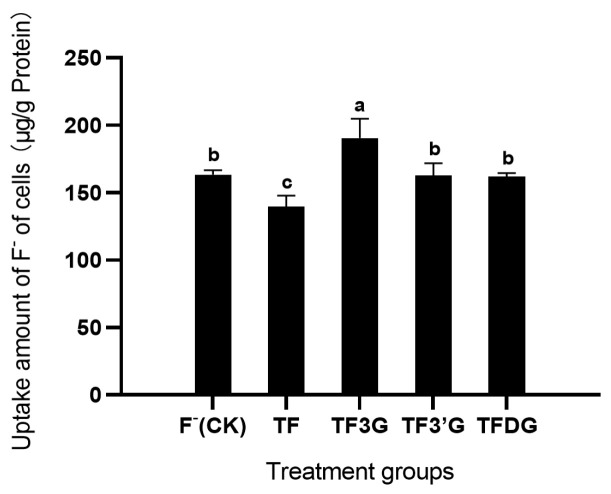
Uptake of F^−^ in Caco-2 cells under different TFs treatments for 2 h. Different lowercase letters in the same group indicate significant differences at *p* < 0.05.

**Figure 5 foods-12-01487-f005:**
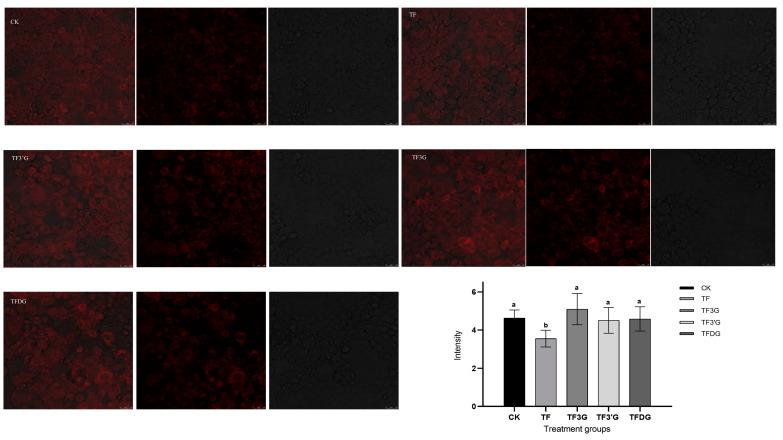
F^−^ fluorescence localization and mean fluorescence intensity of Caco-2 cells in different treatment groups. Each group photos from left to right were the overlap field, fluorescence field, and light field, respectively. Different lowercase letters in the same group indicate significant differences at *p* < 0.05.

**Figure 6 foods-12-01487-f006:**
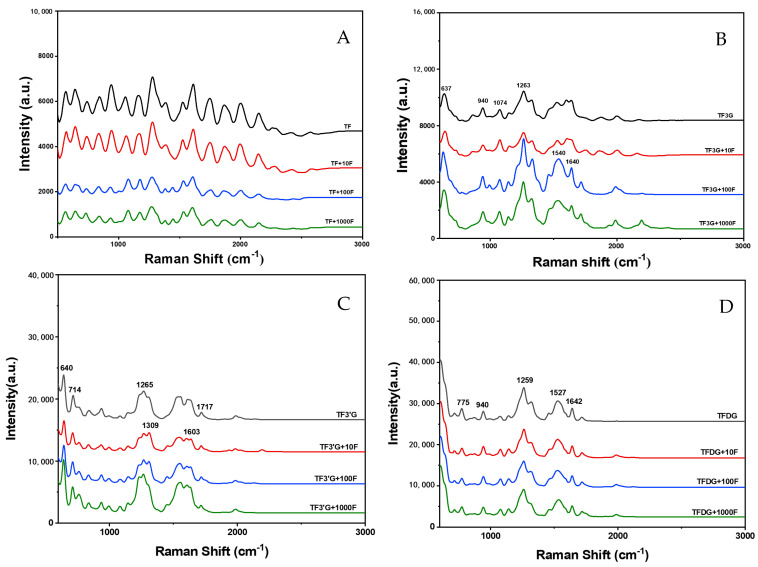
Raman spectra of TFs + different concentrations of F^−^: (**A**) TF treatment group, (**B**) TF3G treatment group, (**C**) TF3′G treatment group, and (**D**) TFDG treatment group.

**Figure 7 foods-12-01487-f007:**
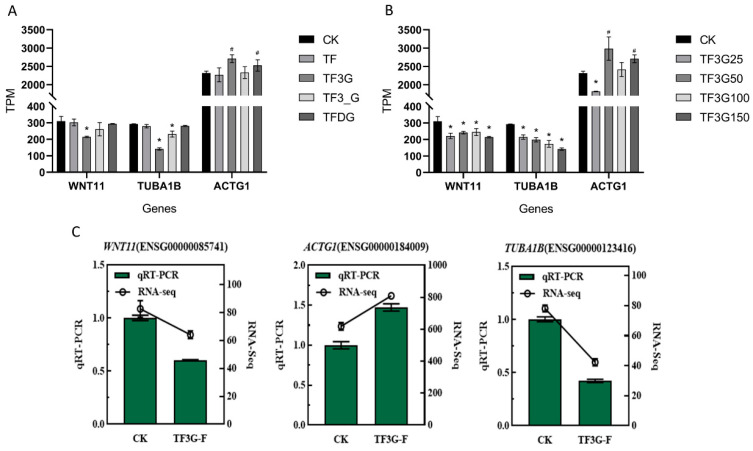
(**A**) Expression of key DEGs under different TFs treatments. (**B**) Expression of key DEGs under the treatment of different concentrations of TF3G. (**C**) qRT-PCR analysis of key DEGs between TF3G and CK groups. ***,** significant decrease relative to CK at *p* < 0.05; # significant increase relative to CK at *p* < 0.05.

**Figure 8 foods-12-01487-f008:**
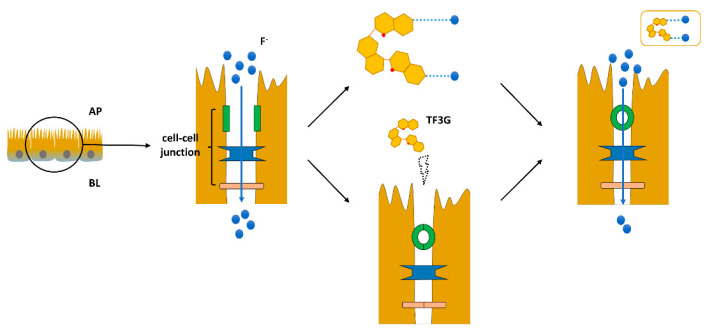
Schematic diagram of TF3G regulating intestinal fluoride absorption.

**Table 1 foods-12-01487-t001:** Effects of TFs on the apparent permeability coefficient of F^−^ transport in Caco-2 cells at different treatment time points.

Time/h	0.5			1			2			3			4		
	Papp (×10^−6^ cm/s)		P-Ratio	Papp (×10^−6^ cm/s)		P-Ratio	Papp (×10^−6^ cm/s)		P-Ratio	Papp (×10^−6^ cm/s)		P-Ratio	Papp (×10^−6^ cm/s)		P-Ratio
	AP-BL	BL-AP		AP-BL	BL-AP		AP-BL	BL-AP		AP-BL	BL-AP		AP-BL	BL-AP	
F(CK)	27.64 ± 1.95 e	29.59 ± 2.09 d	1.07 ± 0.01 b	26.02 ± 1.96 b	31.71 ± 1.60 b	1.22 ± 0.03 c	30.85 ± 1.15 a	31.0 ± 0.60 a	1.01 ± 0.02 b	23.59 ± 0.56 a	26.05 ± 0.38 a	1.11 ± 0.02 b	18.07 ± 0.83 a	20.59 ± 0.82 a	1.14 ± 0.03 b
TF + F	58.23 ± 2.08 a	46.77 ± 5.09 b	0.80 ± 0.06 c	43.27 ± 2.54 a	35.74 ± 2.57 ab	0.83 ± 0.03 d	23.39 ± 1.40 c	24.22 ± 1.26 c	1.04 ± 0.01 b	21.52 ± 1.00 b	23.60 ± 0.94 b	1.10 ± 0.04 b	16.11 ± 0.46 b	18.13 ± 0.31 b	1.13 ± 0.01 b
TF3G + F	39.79 ± 5.43 c	42.12 ± 0.50 bc	1.07 ± 0.13 b	23.51 ± 2.57 bc	33.45 ± 5.92 ab	1.56 ± 0.15 b	20.31 ± 1.82 c	28.04 ± 2.25 b	1.39 ± 0.13 a	18.12 ± 0.84 c	20.76 ± 0.86 c	1.15 ± 0.01 b	13.89 ± 0.77 c	17.35 ± 1.37 b	1.25 ± 0.03 a
TF3′G + F	36.76 ± 0.65 c	49.62 ± 3.40 ab	1.35 ± 0.07 a	20.62 ± 1.08 c	39.09 ± 3.01 a	1.90 ± 0.05 a	23.58 ± 1.46 c	26.13 ± 1.25 b	1.11 ± 0.02 b	18.58 ± 0.82 c	23.47 ± 0.91 b	1.26 ± 0.02 a	14.70 ± 0.06 c	18.16 ± 1.25 b	1.24 ± 0.08 a
TFDG + F	48.86 ± 1.91 b	54.18 ± 2.05 a	1.11 ± 0.01 b	44.31 ± 5.67 a	32.63 ± 4.86 b	0.74 ± 0.01 d	27.03 ± 2.55 b	27.45 ± 0.74 b	1.02 ± 0.07 b	21.06 ± 0.59 b	22.98 ± 1.11 b	1.09 ± 0.02 b	17.11 ± 0.81 ab	17.52 ± 1.06 b	1.02 ± 0.03 c

Note: F: 5 μg/mL F^−^, TFs + F^−^: 150 μg/mL TFs + 5 μg/mL F^−^. Different lowercase letters in the same column indicate significant differences at *p* < 0.05.

**Table 2 foods-12-01487-t002:** Effects of different concentrations of TFs on the apparent permeability coefficient of F^−^ transport in Caco-2 cells.

Concentration (μg/mL)	25			50			100			150		
	Papp (×10^−6^ cm/s)		P-Ratio	Papp (×10^−6^ cm/s)		P-Ratio	Papp (×10^−6^ cm/s)		P-Ratio	Papp (×10^−6^ cm/s)		P-Ratio
	AP-BL	BL-AP		AP-BL	BL-AP		AP-BL	BL-AP		AP-BL	BL-AP	
F(CK)	29.41 ± 0.65 b	30.34 ± 0.59 a	1.02 ± 0.01 a	29.41 ± 0.65 a	30.34 ± 0.59 ab	1.02 ± 0.01 a	29.41 ± 0.65 a	30.34 ± 0.59 a	1.02 ± 0.01 ab	29.41 ± 0.65 a	30.34 ± 0.59 a	1.02 ± 0.01 b
TF + F	31.71 ± 0.65 ab	28.08 ± 4.32 ab	0.88 ± 0.12 b	29.18 ± 0.65 ab	26.78 ± 0.63 cd	0.93 ± 0.01 a	26.94 ± 0.85 a	26.66 ± 1.63 bc	0.99 ± 0.03 b	22.72 ± 1.37 c	23.68 ± 1.24 c	1.04 ± 0.01 b
TF3G + F	28.43 ± 0.71 bc	26.27 ± 0.22 b	0.91 ± 0.00 b	26.87 ± 1.35 b	24.13 ± 1.43 de	0.92 ± 0.03 a	26.81 ± 1.25 a	28.10 ± 1.02 ab	1.08 ± 0.01 a	19.71 ± 1.78 c	27.44 ± 2.22 b	1.40 ± 0.13 a
TF3′G + F	28.89 ± 0.08 c	25.99 ± 1.94 b	0.93 ± 0.04 ab	25.94 ± 1.43 b	21.82 ± 0.54 e	0.86 ± 0.02 a	25.82 ± 0.28 b	24.67 ± 1.38 c	0.96 ± 0.04 b	22.90 ± 1.42 c	25.55 ± 1.23 bc	1.12 ± 0.02 b
TFDG + F	32.67 ± 1.55 a	30.10 ± 0.01 a	0.92 ± 0.04 b	30.51 ± 3.48 a	30.17 ± 1.09 bc	0.99 ± 0.08 a	26.64 ± 4.07 a	28.04 ± 0.78 bc	0.94 ± 0.07 b	26.27 ± 2.49 b	26.85 ± 0.72 b	1.03 ± 0.07 b

Note: F: 5 μg/mL F^−^, treatment time: 2 h; Different lowercase letters in the same column indicate significant differences at *p* < 0.05.

**Table 3 foods-12-01487-t003:** Effects of different concentrations of F^−^ on the apparent permeability coefficient of F^−^ transport in Caco-2 cells.

F− Concentration (μg/mL)	0.2			1			2			5		
	Papp (×10^−6^ cm/s)		P-Ratio	Papp (×10^−6^ cm/s)		P-Ratio	Papp (×10^−6^ cm/s)		P-Ratio	Papp (×10^−6^ cm/s)		P-Ratio
	AP-BL	BL-AP		AP-BL	BL-AP		AP-BL	BL-AP		AP-BL	BL-AP	
F(CK)	23.26 ± 2.33 a	11.22 ± 0.75 a	0.53 ± 0.03 c	11.90 ± 0.07 a	12.05 ± 0.78 a	1.01 ± 0.02 c	10.52 ± 1.50 a	14.61 ± 1.57 a	1.29 ± 0.14 b	29.41 ± 0.65 a	30.34 ± 0.59 a	1.02 ± 0.01 b
TF + F	18.36 ± 1.22 b	11.29 ± 0.84 a	0.62 ± 0.01 ab	8.05 ± 0.36 b	10.35 ± 0.31 b	1.29 ± 0.02 a	7.88 ± 1.03 b	10.70 ± 0.30 b	1.37 ± 0.16 b	22.72 ± 1.37 c	23.68 ± 1.24 c	1.04 ± 0.01 b
TF3G + F	17.22 ± 1.56 bc	10.70 ± 0.07 ab	0.62 ± 0.05 a	9.01 ± 0.58 b	10.39 ± 0.73 b	1.15 ± 0.01 b	8.13 ± 0.38 b	13.40 ± 0.69 a	1.65 ± 0.03 a	19.71 ± 1.78 c	27.44 ± 2.22 B	1.40 ± 0.13 a
TF3′G + F	14.37 ± 0.96 c	8.06 ± 0.28 b	0.56 ± 0.02 bc	5.92 ± 0.61 c	7.28 ± 0.47 c	1.23 ± 0.08 a	6.82 ± 0.09 bc	7.55 ± 0.58 c	1.11 ± 0.08 c	22.90 ± 1.42 c	25.55 ± 1.23 bc	1.12 ± 0.02 b
TFDG + F	14.18 ± 0.83 c	7.63 ± 0.08 b	0.54 ± 0.03 c	6.70 ± 0.28 c	7.22 ± 0.34 c	1.08 ± 0.01 c	5.62 ± 0.43 c	8.29 ± 0.23 c	1.48 ± 0.07 ab	26.27 ± 2.49 b	26.85 ± 0.72 b	1.03 ± 0.07 b

Note: TFs concentration: 150 μg/mL, treating time: 2 h. Different lowercase letters in the same column indicate significant differences at *p* < 0.05.

**Table 4 foods-12-01487-t004:** Thermodynamic parameters of interaction between F^−^ and TFs.

	F^−^-TF	F^−^-TF3G	F^−^-TF3′G	F^−^-TFDG
Ka (1/M)	2.014 × 10^2^	1.854 × 10^3^	3.262 × 10^1^	1.732 × 10^3^
ΔH (kJ/mol)	266.3	−5.251	102.5	13.44
n	0.023	0.744	0.866	0.645
Kd (M)	4.965 × 10^−3^	5.395 × 10^−4^	3.066 × 10^−2^	5.772 × 10^−4^
ΔS (J/mol⋅K)	937.3	44.95	372.7	107.1
ΔG (kJ/mol)	−13.156	−18.653	−8.621	−18.492

Note: ΔG = ΔH − TΔS, T = 298.15 K.

## Data Availability

Data are contained within the article or Appendix A.

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
