# Peer review of "Effect and Mechanism of Theaflavins on Fluoride Transport and Absorption in Caco-2 Cells"

_foods, 2023, doi:10.3390/foods12071487_

Round 1

Reviewer 1 Report

The manuscript describes the effect and mechanism of theaflavins

(TFs) on fluoride (F-) uptake and transport in the Caco-2 cell model through structural chemistry and transcriptome analysis. The topic is relevant to the aim and scope of the Foods in terms of both the end applications and the physiological materials used. The manuscript is well-written and easy to follow. Some clarifications in the texts are needed. Overall, this manuscript needs the below comments:

1)      Effect of TFs and F- treatment on the viability of Caco-2 cells was investigated respectively for 24 hours, as shown in Figure 2. However, why was the effect of the TFs + F- mixtures only investigated for 4 hours instead of 24 hours?

2)      Why did TFDG+F- have less effect rather than the others?

3)      It is not clear to recognize the mechanism of the fluoride transport. Is the transport affected by the fluoride itself or through the fluoride accumulation?

4)      Did the results shown in Figure 4 have a trend identical to those in Figure 3? Especially TF3G?

5)      As a thermodynamic variable, Gibb’s free energy is most commonly used. How about Gibb’s free energy for the interaction between F- and TFs?

6)      What was the evidence to support the direct binding to F-?

Reviewer 2 Report

The manuscript seems interesting, but I have a number of comments.

In my opinion, the Introduction could be expanded. Statistical data on the prevalence of fluoride deficiency in the population would be of interest, and data on the association (if any) of tea consumption with fluoride deficiency is also needed.

Figure 2 does not indicate the statistical significance of differences between the data, although they are observed. This needs to be corrected.

I can also recommend regrouping the data in Figure 2a. Group them (along the x-axis) not by concentration, but by substance. However, this is only advice.

Figure 5 raises many questions. If the goal is to show the localization of fluoride ions in cells, then the transmitted light image, as well as the specific staining of the organelles (nucleus, membranes) should be given. As it stands, the images are completely unreadable and cannot be analyzed.

What fluoride ion Probe1 are you talking about? Since the study must be reproducible and verifiable, the authors should indicate which fluorescent dye is in question. If it is a self-prepared molecule, then its structure and spectral characteristics should be given.

The conclusion can also be improved by analyzing the impact of the data obtained in the context of tea consumption in the future or in the current, if there are some medical statistics on tea consumption and the level of fluoride in the human body.

Round 2

Reviewer 1 Report

The issues have been addressed except one thing, which includes the description of the response 5.

Reviewer 2 Report

Dear authors,

Great thanks for your reply and careful corrections according to meine and other reviewer's comments.

Your manuscript became much better, and now I can accept it in present form.

Author Response

Thank you again for your comments and suggestions for our manuscript.